# Intracranial Germinoma—Association between Delayed Diagnosis, Altered Clinical Manifestations, and Prognosis

**DOI:** 10.3390/cancers15102789

**Published:** 2023-05-17

**Authors:** Iwona Jabłońska, Marcin Goławski, Elżbieta Nowicka, Katarzyna Drosik-Rutowicz, Anna Trybus, Rafał Tarnawski, Marcin Miszczyk

**Affiliations:** 1IIIrd Department of Radiotherapy and Chemotherapy, Maria Sklodowska-Curie National Research Institute of Oncology, 44-102 Gliwice, Poland; 2Department of Pharmacology, Faculty of Medical Sciences in Zabrze, Medical University of Silesia, 40-055 Katowice, Poland; 3Ist Department of Radiotherapy and Chemotherapy, Maria Sklodowska-Curie National Research Institute of Oncology, 44-102 Gliwice, Poland

**Keywords:** intracranial germinoma, clinical manifestation, delayed diagnosis, germ cell tumour

## Abstract

**Simple Summary:**

Intracranial germinoma is a rare tumour of the nervous system, primarily affecting children and young adults, causing neurological and endocrinological symptoms. In this study, we found that a delayed diagnosis (>6 months after initial symptoms) leads to worse overall survival rate despite smaller sizes of the primary tumours. The time to diagnosis was shorter in patients who exhibited neurological symptoms, and bifocal lesions were associated with a significantly worse prognosis. This study emphasizes the need for timely diagnosis of intracranial germinoma, and suggests the necessity of thorough differential diagnosis in patients presenting atypical neurological and endocrinological symptoms.

**Abstract:**

Background: Intracranial germinoma is a rare malignant neoplasm of the central nervous system (CNS) that occurs in children and young adults. The aim of our study was to assess the initial manifestation of the disease, and to find differences in outcomes dependent on time of diagnosis. Methods: The study group consisted of 35 consecutive patients (adults and children) who were treated for intracranial germinoma with radiotherapy at a tertiary centre, and their data were retrospectively collected. We evaluated time from the first symptoms to diagnosis and divided patients into early and delayed diagnosis groups. Delayed diagnosis has been defined as the time from initial presentation to final diagnosis longer than six months. Results: A total of 17 (48.6%) of the patients had delayed diagnoses. Patient survival data spanned a median of six (interquartile range 3–12) years. At the time of the diagnosis, patients presented exclusively neurological symptoms in 16 (45.7%) cases, exclusively endocrinological symptoms in five (14.3%) cases, and mixed symptoms in the remaining cases (*n* = 14; 40.0%). Patients with neurological symptoms had shorter time (*p* < 0.001) from first symptoms to the final diagnosis (5.91 months) than in patients without them (19.44 months). The delayed diagnosis group presented significantly smaller tumour size (mean maximal dimension 2.35 cm) compared to early diagnosis group (3.1 cm). The 5-year and 10-year survival rates of our patients were 94.3% and 83.4%, respectively. Patients with a delayed diagnosis (*n* = 17) had a significantly worse (*p* = 0.02) 10-year OS (63%) compared to the early diagnosis group (*n* = 18; OS = 100%). Importantly, in five patients (14.29%), initial manifestation occurred before radiological signs of the disease. Conclusion: Our study stresses the need for timely diagnosis in intracranial germinoma, as a delay has a significant impact on the prognosis. In particular, if the tumour is small or causes exclusively endocrinological symptoms, the diagnosis may be difficult and delayed.

## 1. Introduction

Intracranial germinoma is a rare malignant neoplasm of the central nervous system (CNS) that occurs in children and young adolescents [1]. It belongs to the germ cell tumours (GCT) family, a heterogeneous group of neoplasms that share a presumed common origin with primordial germ cells (PGC) [2,3]. These tumours can be divided into teratomas, germinomatous (GGCT), and nongerminomatous germ cell tumours (NGGCT). However, their histological classification remains controversial [4].

GCTs occur predominantly in testes and ovaries, but can also appear in extragonadal locations along the midline of the body. They are associated with the migration of PGCs from the yolk sac along the dorsal mesentery of the hindgut to the genital ridges during early development [5,6,7]. GGCTs and NGGCTs can also be found in two extragonadal sites: the anterior mediastinum and the brain [5]. Intracranial germinoma accounts for around two-thirds of CNS GCTs [1], and it is located mainly on the midline, especially in the pineal or suprasellar region [8]. In approximately 2–26% of intracranial cases, the tumour occurs in both regions as bifocal germinoma [9]. Additionally, germinoma can uncommonly be found in extraaxial sites such as the basal ganglia, thalamus, corpus callosum, ventricles [8,9,10], or even optic nerve [11] and cerebellar hemisphere [12]. Moreover, intracranial germinomas can rarely be diagnosed at the stage of disseminated disease.

The primary site of the tumour, as well as its size, are associated with its clinical manifestation. Initial symptoms may include neurological deficits, such as motor and visual impairment, which are often associated with hydrocephalus [8,13]. Furthermore, patients often present with endocrinological symptoms, such as diabetes insipidus (DI), growth deficiency, central hypothyroidism, secondary hypoadrenocorticism, and hypogonadism [8,14], caused by the infiltration of the pituitary gland.

The overall incidence of CNS GCTs is approximately 0.6 cases per million in the US, and 1 case per million in the United Kingdom and Germany. The incidence of CNS GCTs is higher in Asia compared to the United States and European countries. For example, a higher occurrence was observed in Japan (2.7 per million) [15]. Intracranial germinoma most frequently occurs in children and adolescents, with 35–40% of cases diagnosed before the age of 14, and 90% before the age of 20 [16]. The median age of the diagnosis is 16 years in the United States [17].

The survival rates are exceptionally high, with as much as 97% five-year event-free survival in patients treated with craniospinal axis irradiation [9]. Considering the excellent oncological outcomes in terms of survival, the side effects of the treatment remain a valid concern. Some studies have suggested that limiting radiotherapy fields leads to decreased treatment toxicity [18,19,20]. However, treating germinoma with chemotherapy alone resulted in less than 50% of cases being cured [21] and even chemotherapy with local radiotherapy still resulted in more relapses than craniospinal irradiation without chemotherapy [22]. Therefore, there is an ongoing debate on the optimal treatment modality, which would ideally balance high cure rates and possible side effects of radiotherapy. The recently published guidelines by EANO suggest that an optimum combination of chemo- and radiotherapy should be considered as the treatment of choice—adjuvant chemotherapy combined with reduced dose and field radiotherapy are now regarded as the standard of care in case of localized tumours [9].

Considering the limited data on this subject, we described our institutional experience with the treatment of intracranial germinoma. Regarding recent guidelines, the article is focused on the initial clinical manifestations of the disease and differences in outcomes dependent on the time of diagnosis.

## 2. Materials and Methods

### 2.1. Patient Characteristics

The study group consisted of 35 consecutive patients (adults and children) who were treated for intracranial germinoma with radiotherapy at a tertiary centre: Maria Sklodowska-Curie National Research Institute of Oncology, Gliwice, Poland, from January 2000 to August 2021. Patients with NGGCT were excluded from the study, as well as those with previous CNS tumours or other neoplasms.

The data were retrospectively collected and patient’s age at diagnosis, sex, initial symptoms, and comorbidities were analysed. Furthermore, we assessed treatment outcomes including 5-year and 10-year Overall Survival (OS) and Progression-Free Survival (PFS). The OS and PFS were measured from the day of diagnosis to death for OS, and to radiologically confirmed progression or recurrence of the disease or death for PFS. For the purposes of survival estimations, data from National Cancer Register were used in addition to data collected within our institution.

Moreover, we evaluated the time from the first symptoms to diagnosis and divided patients into early and delayed diagnosis groups. We defined delayed diagnosis as the time from initial presentation to final diagnosis longer than six months. If available, the data on laboratory blood tests (including AFP and beta-HCG levels) and tumour immunohistochemical markers were collected. AFP elevation was defined as level above 10 ng/mL and beta-HCG elevation was defined as level above 3 IU/L. Radiological findings (computed tomography [CT] and magnetic resonance imaging [MRI]) were recorded with regard to tumour location, size, dissemination of the disease, and presence of hydrocephalus. We defined the disseminated disease as the presence of multifocal lesions observed in MRI before the histopathological confirmation. Tumour size was assessed by measuring its maximal dimension and volume. If patients had more than one lesion, only the maximal dimension of the largest lesion has been measured. Tumour volume was calculated using the formula for the volume of an ellipsoid.

### 2.2. Statistical Analysis

Kaplan-Meier method and log-rank tests were used for OS and PFS comparisons and estimations. Receiver-Operator Curve analysis was used to determine cut-offs for total tumour volume. Due to small sample size, bimodal variables were analysed using Fisher exact test or, when any of the expected values was larger than five, Yates’s chi-squared test. Freeman-Halton extension of the Fisher exact test was used for trimodal variables. Quantitative variables were analysed using the Student *t*-test, unless the distribution was not normal, in which case the values were log-transformed or the Mann-Whitney test was used. The results are presented as median and interquartile range (IQR) or percentages. The statistical analysis was performed using the STATISTICA 13.1 software with Medical Bundle (StatSoft INC., Tulsa, OK, USA).

## 3. Results

The median age at diagnosis was 16 years (IQR 13–22 years), 19 (54.3%) patients were below 18 years at diagnosis, and only 7 (20%) patients were female. According to the previously defined criteria, 17 (48.6%) of the patients had delayed diagnoses. Median medical history-based follow-up was 35 months (IQR 1–72 months), and the median follow-up for census-based survival data was 71 months (IQR 33–145 months).

Thirty-three patients had histopathologically confirmed pure germinoma. The two remaining cases were diagnosed as germinoma based on the typical bifocal localization of the disease and clinical characteristics.

Tumour localization in the pineal gland area was the most common in our study (*n* = 11; 31.4%), although there were differences between groups with early and delayed diagnosis. Suprasellar and bifocal localizations of the tumour were the most common in patients with delayed diagnosis (both *n* = 6; 35.3%), whereas in patients with early diagnosis the tumour was located in the pineal gland in most cases (*n* = 8; 44.4%). Three patients were diagnosed with disseminated disease, and in one case dissemination of bifocal germinoma occurred before the start of the treatment, but after histopathological confirmation. A comparison of clinical and radiologic findings between the groups with early and delayed diagnosis is presented in Table 1.

The median tumour size (defined as the largest dimension of the largest lesion) in our patients was 2.70 cm (IQR 2–3.8 cm). However, it was significantly greater (*p* = 0.04) in patients with early diagnosis (3.1 cm; IQR 2.5–4.0 cm), compared to the delayed diagnosis group (2.35 cm; range 1.9–3.2 cm). There was no significant difference between total tumour volume between groups (*p* = 0.174) but the difference in the size of the largest tumour approached statistical significance (*p* = 0.051). Hydrocephalus was found in 57.1% of cases, and there were no significant differences between the early and delayed diagnosis groups in this regard.

At the time of diagnosis, our patients presented exclusively neurological symptoms in 16 (45.7%) cases, exclusively endocrinological symptoms in 5 (14.3%) cases, and mixed symptoms in the remaining cases (40.0%; *n* = 14). Detailed data about the initial manifestation of the disease are shown in Table 2. Ocular manifestations were the most common group of symptoms (68.6%; *n* = 24). Diplopia was the most frequently occurring type of visual disturbance and it occurred in 11 patients (31.4%).

The average time from first symptoms to the final diagnosis was significantly lower (*p* < 0.001; Student *t*-test of log value) in patients with neurological symptoms (5.91 months) than in patients without them (19.44 months). Hydrocephalus occurred in 63.3% of the patients with neurological symptoms.

Tumours in the suprasellar and bifocal areas were significantly associated with the occurrence of endocrinological symptoms (*p* = 0.047 and *p* = 0.022, respectively, Fisher exact test). On the other hand, tumour location in the pineal gland was significantly (*p* = 0.011; Yate’s chi-squared test) more frequent in patients without endocrinological symptoms. The majority of the differences in symptoms did not differ significantly between groups, although combined pituitary hormone deficiency (*p* = 0.032), change of behaviour (*p* = 0.007), and especially lethargy (*p* = 0.045) were significantly more frequent in the delayed diagnosis group. The early diagnosis group presented a headache more often than the late diagnosis group, but the difference was not statistically significant (*p* = 0.062).

All of our patients received radiotherapy. Most of the patients (*n* = 21; 60%) received whole ventricular irradiation. A total of twelve patients underwent craniospinal irradiation, and three patients had whole brain irradiation. Additionally, 32 patients received a boost of radiotherapy to the tumour or tumour bed. Radiotherapy was preceded by chemotherapy in 74.2% (*n* = 26) of cases. Furthermore, 37.1% (*n* = 13) of patients underwent surgery before chemotherapy (22.8%, *n* = 8) or radiotherapy (14.3%, *n* = 5), and four patients received radiotherapy as a single treatment method.

A total of 24 (80%) patients attended follow-up visits at the study centre for an average of 6.29 years. Our patients had very good outcomes, with 5-year and 10-year survival rates of 94.3% and 83.4%, respectively. There was a difference between the early and delayed diagnosis groups. Patients with a delayed diagnosis (*n* = 17) had a significantly worse (*p* = 0.020) 10-year OS (63%) compared to the early diagnosis group (*n* = 18, OS = 100%). Although 5- and 10-year progression-free survival rates were better in the early diagnosis group (84.3% for both 5 and 10 years) than in the delayed diagnosis group (61.8% and 61.8%, respectively), the difference was not statistically significant (*p* = 0.205). Kaplan-Meier curves for the OS rate of patients with an early and delayed diagnosis are presented in Figure 1. Two patients died due to the progression of the disease, and in the remaining two cases, death was caused by the pulmonary and ventricular embolism during the treatment. Of note, bifocal location of the lesions was associated with worse prognosis (*p* = 0.033) compared to other locations (Figure 2).

ROC analysis was undertaken to determine cut-offs for total tumour volume and largest tumour volume for the purposes of survival analysis. The result was 7.720 cm^2^ for total tumour volume and 4.984 cm^2^ for the volume of the largest tumour. Total tumour volume above this size was interestingly associated with improved prognosis (*p* = 0.046; Figure 3), and for the largest tumour volume, the association was even stronger (*p* = 0.007; Figure 4).

In five of our patients (14.29%) first symptoms of the disease occurred before a tumour was detectable in MRI. In three cases the first MR did not show any abnormalities and in two cases slight pituitary enlargement has been initially detected and was misdiagnosed as lymphocytic hypophysitis. The time from the first symptoms to the final diagnosis was particularly long, and it ranged from eight months to over two years. In two patients the initial manifestation included combined pituitary hormone deficiency with growth retardation as well as diabetes insipidus, and the tumour was eventually detected in suprasellar localization. In another two cases the final diagnosis was bifocal germinoma. One of those patients presented combined pituitary hormone deficiency; however, this was without growth retardation. The second patient reported only headache and visual impairment. In the last case, the tumour was finally found in the pituitary gland, and the patient reported visual impairment as the first disease symptom. Before the treatment, total tumour volume in those difficult-to-diagnose patients ranged from 0.29 cm^3^ to 2.96 cm^3^. Unfortunately, one patient without radiologically detectable lesion at the beginning of the disease has died during treatment.

## 4. Discussion

This study analysed the initial manifestation and outcomes of 35 patients with intracranial germinoma. Our patients had 5-year and 10-year survival rates of 94.3% and 83.4%, respectively. Other authors also present excellent outcomes. In a retrospective analysis conducted by Kang et al., 5-year and 10-year OS rates were 100% and 89.5%, respectively. The study involved 29 adult patients and 67 pediatric patients. Early and delayed diagnosis groups were not compared, although children were demonstrated to have a significantly better progression-free survival and overall survival. Such a trend was also seen in our group, but the difference was not significant (*p* = 0.205). An even more significant risk factor was the presence of bifocal lesions. This finding was replicated in our study (*p* = 0.033) [23]. Of note, bifocal lesions were associated with worse prognosis in all suprasellar germ cell tumours [24]. Frappaz et al. reported 100% overall survival after a median follow-up of 41 months (ranging from 4 to 194 months) [8]. The 5-year OS reported for intracranial germinoma by Paximadis et al. is 85.7% [25]. Another study reported 5- and 10-year OS of 94.5% and 91.3%, respectively [26]. In a study of 189 patients aged 4–47, 10-year OS of 85.2% or 92.8% was reported, dependent on the use of chemotherapy, which increased survival rate [18]. What is most important, in our study 10-year OS in patients with delayed diagnosis was found to be 63%, compared with 100% in the early diagnosis group. An interesting finding is that the group with delayed diagnosis had smaller tumour sizes measured by a maximal dimension and the largest tumour volume. A possible explanation is that a smaller tumour may be more difficult to accurately diagnose or may cause more subtle and difficult-to-interpret symptoms. Despite smaller tumour size in the delayed diagnosis group, those patients displayed significantly worse OS. In our study, five patients (14.29% of whole group) were particularly difficult to diagnose, because they presented the first symptoms of the disease a couple of months before the tumour was detectable in MRI. It is a very interesting finding, because brain tumours can usually grow without any symptoms for a prolonged period of time. To the contrary, in some of our patients the symptoms occurred before the tumour was even detectable. In our study, patients with total tumour volume greater than 7.72 cm^3^ and largest tumour volume greater than 4.98 cm^3^ had significantly better OS (*p* = 0.046 and *p* = 0.007, respectively). This is another interesting finding that requires further research.

To our knowledge, this is the first study that assessed the difference in the OS and tumour size between groups of patients with early and delayed diagnosis of intracranial germinoma. It also appears to be the first study involving a large group of patients, in whom the first neurological and endocrinological symptoms occurred before the tumour could be diagnosed by the MRI [27,28].

Intracranial germinoma causes various neurological and endocrinological signs and symptoms [29]. Neurological manifestations include ocular complaints due to the typical localization of the tumour in the suprasellar area near the optic chiasm [30]. Other neurological symptoms such as headaches, nausea, or vomiting are related to increased intracranial pressure. Endocrinological manifestations include growth deficiency, hyperprolactinemia, adrenal axis dysfunction, gonadal axis dysfunction, diabetes insipidus, and combined pituitary hormone deficiency, and they are thought to be caused by the involvement of the suprasellar and hypothalamic regions [31,32]. In our study, endocrinological manifestation was also associated with suprasellar and bifocal tumour location, while patients without endocrinological symptoms more frequently had tumours in the pineal gland. Patients with exclusively endocrinological symptoms presented significantly higher time from first symptoms to the final diagnosis (19.44 months), compared to patients with neurological symptoms (5.91 months). When we divided patients into groups with early and delayed diagnosis, patients in the delayed diagnosis group presented combined pituitary insufficiency more often. Similar results were reported by Chang et al., who conducted a retrospective study of 49 patients with pure intracranial germinoma focusing solely on children. They divided patients into groups with early (*n* = 37; 75.5%) and delayed (*n* = 12; 24.5%) diagnoses. The delayed diagnosis was defined as time to diagnosis longer than six months from the patient’s initial presentation. The authors found that patients in the early diagnosis group suffered from initial neurological manifestation significantly more often than the delayed diagnosis group, whereas patients with delayed diagnosis were more likely to present endocrinological symptoms [13]. In our study, the percentage of patients with delayed diagnosis (*n* = 17; 48.6%) was significantly higher, possibly due to the less common occurrence of germinomas in Europe and subsequently more difficult diagnostic processes. Chang et al. noted that the delayed diagnosis group had a higher percentage of female patients: 50.0% in the delayed diagnosis group vs. 16.2% in early diagnosis group; this is similar to our study, in which female patients comprise 29.4% and 11.1% of patients in the respective groups. Mixed manifestations of the disease were more common in our study (40% vs. 12.2% in the study of Chang et al.), whereas neurological manifestation was less common (45.7% vs. 65.3%) [13].

In our patients, visual disturbances was the most common group of symptoms. Frappaz et al. conducted a retrospective study investigating visual complaints in 28 patients with intracranial germinoma. Visual disturbances were noted in 12 (42.8%) patients. They included diplopia, loss of visual acuity, and field defects—similarly to our group. All patients with visual disturbances also had diabetes insipidus—such findings could not be replicated in this study. Here, 14 out of 24 patients with visual complaints had no history of diabetes insipidus [8].

Our study has several limitations. The retrospective character of the study may cause bias. Moreover, the number of patients was limited and therefore the sample size is small. Nevertheless, this study still presents a relatively large group for condition as rare as intracranial germinoma in the European population. Further research is still necessary to determine typical and atypical symptoms and prognostic factors due to the rarity of this condition. Particularly, reports regarding manifestations of this rare disease are needed to better direct and plan diagnostic processes, because early diagnosis leads to a significantly improved outcome.

## 5. Conclusions

Our study stresses the need for a timely diagnosis in intracranial germinoma, as a delay has a significant impact on the prognosis. In particular, if the tumour causes endocrine but not neurological symptoms, the diagnosis may be difficult and delayed. Moreover, in some cases initial radiological findings may not indicate the presence of a tumour—this means that all symptoms patients present must be carefully evaluated. Although intracranial germinoma is a rare condition, doctors must remain aware of intracranial tumours in general in pediatric and adolescent patients who present endocrinological symptoms.

## Figures and Tables

**Figure 1 cancers-15-02789-f001:**
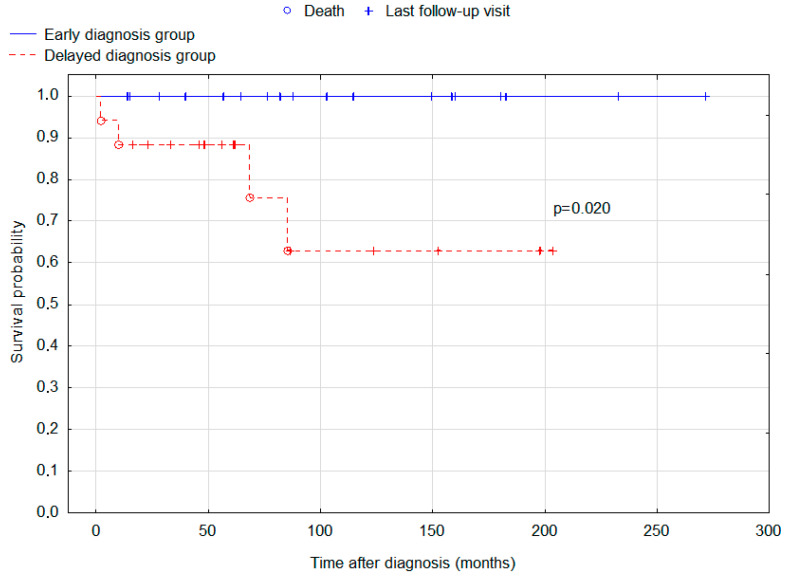
Kaplan-Meier curves for the overall survival of the early and delayed diagnosis groups of patients diagnosed with intracranial germinoma. Log-rank test *p*-value for the difference in survival distributions between the groups = 0.020.

**Figure 2 cancers-15-02789-f002:**
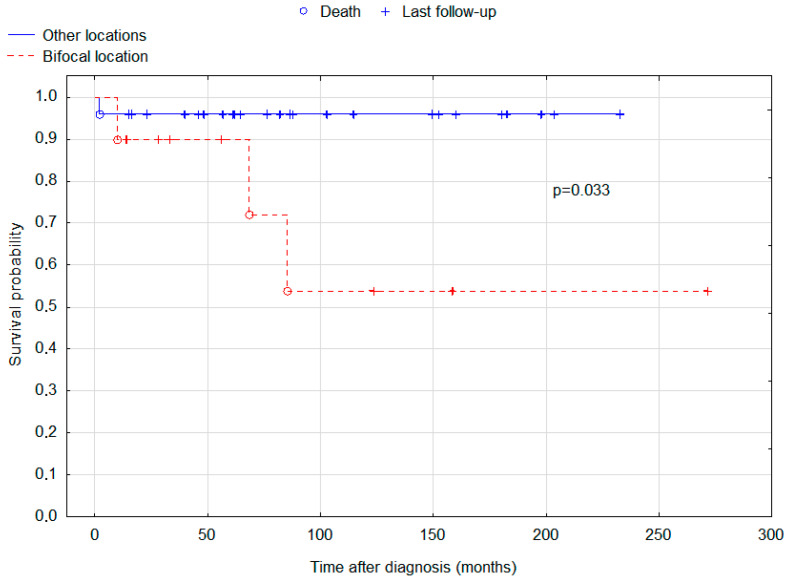
Kaplan-Meier curves for the overall survival of patients depending on tumour location (bifocal versus other locations). Log-rank test *p*-value for the difference in survival distributions between the groups = 0.033.

**Figure 3 cancers-15-02789-f003:**
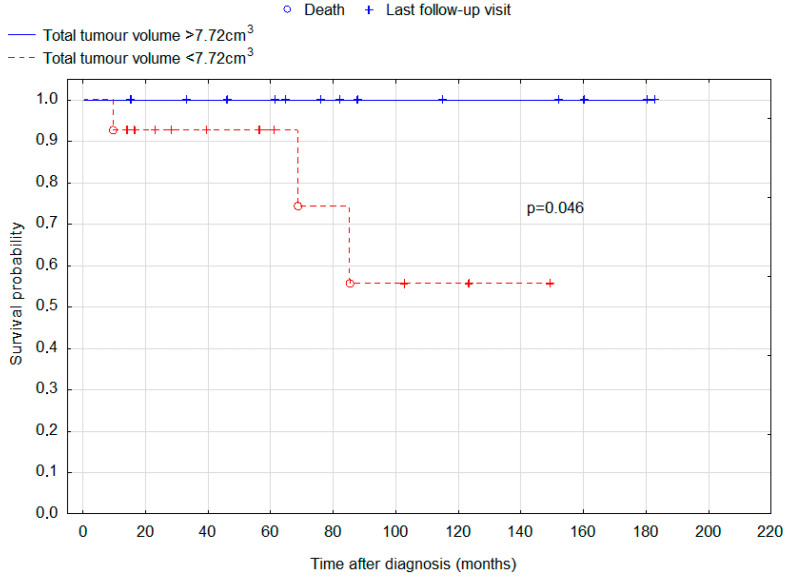
Kaplan-Meier curves for the overall survival of patients depending on total tumour volume. Cut-off between the groups was set to 7.72 cm^3^. Log-rank test *p*-value for the difference in survival distributions between the groups = 0.046.

**Figure 4 cancers-15-02789-f004:**
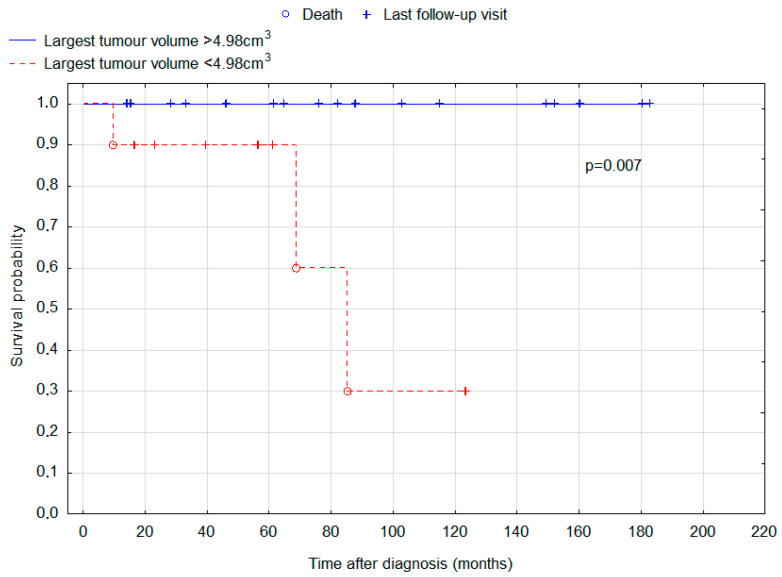
Kaplan-Meier curves for the overall survival of patients depending on the volume of the largest tumour. Cut-off between the groups was set to 4.98 cm^3^. Log-rank test *p*-value for the difference in survival distributions between the groups = 0.007.

**Table 1 cancers-15-02789-t001:** General characteristics of patients. Age and tumour size data are presented as median (interquartile range) and mean (minimum value; maximum value).

	All Patients *n* = 35	Early Diagnosis *n* = 18	Delayed Diagnosis *n* = 17	*p*-Value
**Sex**				0.228
Female	7 (20.0%)	2 (11.1%)	5 (29.4%)	
Male	28 (80.0%)	16 (88.9%)	12 (70.6%)	
**Age (years)**				0.879 *
Median (IQR)	16 (13–22)	17.5 (13–21)	16 (13–22)	
Mean (min; max)	17.84 (5–27)	17.97 (5–26)	17.68 (7–27)	
**Age group**				0.388 †
Children (<18 years old)	19 (54.3%)	8 (44.4%)	11 (64.7%)	
Young adults (>18 years old)	16 (45.7%)	10 (55.6%)	6 (35.3%)	
**Tumour location**				
Pineal gland	11 (31.4%)	8 (44.4%)	3 (17.6%)	0.179 †
Bifocal	10 (28.6%)	4 (22.2%)	6 (35.3%)	0.471
Suprasellar	8 (22.9%)	2 (11.1%)	6 (35.3%)	0.121
Disseminated	3 (8.6%)	2 (11.1%)	1 (5.9%)	1.000
Basal ganglia	3 (8.6%)	2 (11.1%)	1 (5.9%)	1.000
**Maximum tumour dimension**				0.040 ‡
Median (IQR)	2.7 cm (2.0–3.8)	3.1 cm (2.5–4.0)	2.35 cm (1.9–3.2)	
Mean (min; max)	2.85 cm (1.1; 4.5)	3.16 cm (2.0; 4.5)	2.46 cm (1.1; 3.9)	
**Largest tumour volume**				0.051 §
Median (IQR)	6.37 cm^3^ (1.99–13.09)	8.93 cm^3^ (5.85–13.18)	4.03 cm^3^ (1.34–10.97)	
Mean (min; max)	8.64 cm^3^ (0.29; 24.88)	10.36 cm^3^ (17.70; 24.88)	6.49 cm^3^ (0.29; 20.89)	
**Total tumour volume**				0.174 *
Median (IQR)	7.70 cm^3^ (2.96–13.09)	8.93 cm^3^ (5.94–14.26)	4.11 cm^3^ (1.60–10.97)	
Mean (min; max)	8.89 cm^3^ (0.29; 24.88)	10.48 cm^3^ (1.77; 24.88)	6.90 cm^3^ (0.29; 20.89)	
**Hydrocephalus**	20 (57.1%)	11 (61.1%)	9 (52.9%)	0.884 †
Ventriculostomy	10 (28.6%)	5 (27.8%)	5 (29.4%)	0.402
Ventriculoperitoneal shunts	7 (20.0%)	5 (27.8%)	2 (11.8%)	1.000
**Comorbidities**				
Hypertension	2 (5.7%)	0	2 (11.8%)	0.229
Epilepsy	1 (2.9%)	1 (5.6%)	0	1.000
Type 1 diabetes	1 (2.9%)	0	1 (5.9%)	0.486
**Treatment**				
Radiotherapy	35 (100%)	18 (100%)	17 (100%)	1.000
Chemotherapy	26 (74.3%)	13 (72.2%)	13 (76.5%)	1.000
Incomplete excision	9 (25.7%)	4 (22.2%)	5 (29.4%)	0.711
Complete excision	2 (5.7%)	1 (5.6%)	1 (5.9%)	1.000

*p*-values are calculated via a Fisher exact test, unless otherwise indicated. * Student *t*-test. † Yates’s chi-squared test. ‡ Mann-Whitney test. § Student *t*-test of log-transformed values.

**Table 2 cancers-15-02789-t002:** Characteristics of symptoms in patients with early and delayed diagnosis of intracranial germinoma.

	All Patients *n* = 35	Early Diagnosis *n* = 18	Delayed Diagnosis *n* = 17	*p*-Value
**Initial manifestation**				0.282 ‖
Neurological	16 (45.7%)	10 (55.6%)	6 (35.3%)	0.388 †
Endocrinological	5 (14.3%)	1 (5.6%)	4 (23.5%)	0.177
Mixed	14 (40.0%)	7 (38.9%)	7 (41.2%)	0.836 †
**Neurological symptoms**				
Headache	21 (60%)	14 (77.8%)	7 (41.2%)	0.062 †
Nausea/vomiting	10 (28.6%)	5 (27.8%)	5 (29.4%)	1.000
Vertigo and balance disorders	8 (22.9%)	4 (22.2%)	4 (23.5%)	1.000
Drowsiness	5 (14.3%)	2 (11.1%)	3 (17.6%)	0.658
Memory defects	2 (5.7%)	1 (5.6%)	1 (5.9%)	1.000
Hand tremor	1 (2.9%)	0	1 (5.9%)	0.486
**Ocular symptoms**	24 (68.6%)	13 (72.2%)	11 (64.7%)	0.909 †
Diplopia	11 (31.4%)	7 (38.9%)	4 (23.5%)	0.539 †
Blurred vision	5 (14.3%)	2 (11.1%)	3 (17.6%)	0.658
Nystagmus	5 (14.3%)	2 (11.1%)	3 (17.6%)	0.658 †
EOM (extraocular muscle) impairment	3 (8.6%)	0	3 (17.6%)	0.229
Anisocoria	3 (8.6%)	2 (11.1%)	1 (5.9%)	1.000
Visual field defect	3 (8.6%)	2 (11.1%)	1 (5.9%)	1.000
**Motor impairment**				
Facial asymmetry	4 (11.4%)	4 (22.2%)	0	0.104
Hemiparesis	2 (5.7%)	1 (5.6%)	1 (5.9%)	1.000
**Endocrinological symptoms**				
Diabetes insipidus	15 (42.9%)	6 (33.3%)	9 (52.9%)	0.407 †
Combined pituitary Hormone deficiency	13 (37.1%)	3 (16.7%)	10 (58.8%)	0.015 †
Thyroid axis dysfunction	12 (34.3%)	4 (22.2%)	8 (47.1%)	0.234 †
Adrenal axis dysfunction	10 (28.6%)	3 (16.7%)	7 (41.2%)	0.218
Gonadal axis dysfunction	7 (20%)	2 (11.1%)	5 (29.4%)	0.228
Hyperprolactinemia	4 (11.4%)	1 (5.6%)	3 (17.6%)	0.338
Growth deficiency	3 (8.6%)	0	3 (17.6%)	0.104
**Change of behaviour**	6 (17.1%)	0	6 (35.3%)	0.008
**β-HCG elevation**	10/32 (31.3%)	7/16 (43.8%)	3/16 (18.8%)	0.253 †
**AFP elevation**	1/28 (3.7%)	1/16 (6.3%)	0/12 (0%)	1.000

*p*-values are calculated using Fisher exact test unless otherwise indicated. † Yates’s chi-squared test. ‖ The Freeman-Halton extension of the Fisher exact test.

## Data Availability

The datasets used and/or analysed during the current study are available from the corresponding author on reasonable request.

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
