# Peer review of "Intracranial Germinoma—Association between Delayed Diagnosis, Altered Clinical Manifestations, and Prognosis"

_cancers, 2023, doi:10.3390/cancers15102789_

Round 1

Reviewer 1 Report (Previous Reviewer 1)

No further comments

Reviewer 2 Report (Previous Reviewer 3)

After the corrections made after the reviewer`s comments the paper is acceptable for publication in this journal. However, if you have the possibility please do genetic analysis of involved patients with germinoma.

This manuscript is a resubmission of an earlier submission. The following is a list of the peer review reports and author responses from that submission.

Round 1

Reviewer 1 Report

The manuscript was a pleasure to evaluate. The study, analysis and write up was very well done.

Minor suggestions:

Introduction:

- In Line 57 please add in “ 2%-26% of intracranial cases” (if this is what is meant)

Results:

In L138 and L139 please indicate both the numeric value and percentage (n=x; x%) in the brackets.

In L 157 please add “… exclusively with neurological symptoms…”

In Table 1 and 2 please start each line with a capitol letter

Reviewer 2 Report

This is an intersting retrospective study about a possible correlation between prognosis and time between first  examination of symptoms and time of diagnosis in germinomas.

As authors mentions this finding was previously described in 2021 in Pediatrics and Neonatology in Taiwan in a higher cohors of patients (49) with same conclusions.

Major concerns:

The age distribution is a little bit surprising as pure germinoma often occur in patients younger than 13.

Tumor marker elevation (AFP) should not occur in germinomas, therefore it would important to know the rate of elevation or the definition of tumor marker elevation in the study.

The correlation between extent of radiotherapy or application of chemotherapy or extent of disases at the end of tretament and time of diagnosis should be described in the comparing table 

Reviewer 3 Report

The paper describes a respectable number of patients with this rare type of tumor. That, in my opinion, is the only quality of this article. The article should be more comprehensive and with more data than that what is offered in this paper. 

The results of the work are very modest. Analysis of neurological and endocrinological manifestations is inadequate. The given numbers are very small to draw  a statistical conclusion. Such inappropriate statistical analysis demonstrated that the only difference between the early and delayed diagnosis of germinoma is the occurrence of headaches and multiple hormonal deficits. Practically,  the division into these 2 groups did not yield any important conclusions. Due to the rarity of these tumors, my suggestion is to do a genetic analysis of the patient`s tumor (KIT, KRAS, NANOG, NRAS, CCND, BRAF...) by NGS method, and then compare the genotypes and phenotypes of the patients. 

There are no data on tests performed to assess the secretory reserve of the pituitary. It remains an open question on the basis of which the hormone deficit is proven (basal hormone values, ITT, glucagon test...).

Reviewer 4 Report

Thank you for inviting me to review this really lovely and well put together manuscript. Despite it requiring minor English grammar changes it was an absolute pleasure to read and as a first language English speaker I found it understandable and easy to follow. The methods and results section was particularly clearly laid out - thank you. 

Two suggestions:

1. Perhaps add one sentence on the OS and PFS at the end of the results. The figures are clear but do not have any percentages on the curves which would complete the picture.

2. How did or have your findings changed your clinical assessment of patients upfront or the way in which you conduct tumour surveillance. If you answer the "why" of the study this would also round things off nicely.